# Evaluation of the Prognostic Value of Pretherapeutic Magnetic Resonance Imaging in Predicting Soft Tissue Sarcoma Radiation Response: A Retrospective Study from a Large Institutional Sarcoma Imaging Database

**DOI:** 10.3390/cancers16050878

**Published:** 2024-02-22

**Authors:** Guillaume Vogin, Matthias Lepage, Julia Salleron, Mathilde Cuenin, Alain Blum, Pedro Augusto Gondim Teixeira

**Affiliations:** 1Department of Radiation Therapy, Institut de Cancérologie de Lorraine, 6 Avenue de Bourgogne, 54519 Vandœuvre-lès-Nancy, France; m.cuenin@nancy.unicancer.fr; 2Centre François Baclesse, Centre National de Radiothérapie du Luxembourg, BP436, L-4005 Esch-sur-Alzette, Luxembourg; 3UMR 7365 CNRS-UL IMoPA, Biopôle de l’Université de Lorraine, Campus Brabois Santé, 9 Avenue de la Forêt de Haye, BP 20199, 54505 Vandœuvre-lès-Nancy, France; 4Guilloz Imaging Department, University Hospital Center of Nancy, 29 Avenue du Maréchal de Lattre de Tassigny, 54035 Nancy, France; matthlepage@gmail.com (M.L.); a.blum@chru-nancy.fr (A.B.); ped_gt@hotmail.com (P.A.G.T.); 5Biostatistics Unit, Institut de Cancérologie de Lorraine, 6 Avenue de Bourgogne, 54519 Vandœuvre-lès-Nancy, France; jsalleron@gmail.com; 6Université de Lorraine, IADI, Inserm U1254, Bâtiment Recherche CHRU de Nancy Brabois, 5 Rue du Morvan, 54500 Vandœuvre-lès-Nancy, France

**Keywords:** Apparent Diffusion Coefficient (ADC), soft tissue sarcoma, neoadjuvant radiotherapy, Magnetic Resonance Imaging (MRI), posttreatment necrosis

## Abstract

**Simple Summary:**

Radiotherapy is primordial in the local control strategy for soft tissue sarcomas. A complete response to preoperative RT is associated with a gain in survival. It, therefore, appears useful to identify in an early and non-invasive manner which patients will present such a response profile. During a single NETSARC-center retrospective study, we investigated and identified pre-therapeutic multimodal MRI parameters in line with the histological response (fibrosis and necrosis). These parameters must be prospectively validated in the hope of personalizing STS’s local control strategy.

**Abstract:**

**Background:** RT-induced hyalinization/fibrosis was recently evidenced as a significant independent predictor for complete response to neoadjuvant radiotherapy (RT) and survival in patients with soft tissue sarcoma (STS). **Purpose:** Non-invasive predictive markers of histologic response after neoadjuvant RT of STS are expected. **Materials and Methods:** From May 2010 to April 2017, patients with a diagnosis of STS who underwent neoadjuvant RT for limb STS were retrieved from a single center prospective clinical imaging database. Tumor Apparent Diffusion Coefficients (ADC) and areas under the time-intensity perfusion curve (AUC) were compared with the histologic necrosis ratio, fibrosis, and cellularity in post-surgical specimens. **Results:** We retrieved 29 patients. The median ADC value was 134.3 × 10^−3^ mm^2^/s. ADC values positively correlated with the post-treatment tumor necrosis ratio (*p* = 0.013). Median ADC values were lower in patients with less than 50% necrosis and higher in those with more than 50% (120.3 × 10^−3^ mm^2^/s and 202.0 × 10^−3^ mm^2^/s, respectively (*p* = 0.020). ADC values higher than 161 × 10^−3^ mm^2^/s presented a 95% sensitivity and a 55% specificity for the identification of tumors with more than 50% tumor necrosis ratio. Tumor-to-muscle AUC ratios were associated with histologic fibrosis (*p* = 0.036). **Conclusions:** ADC and perfusion AUC correlated, respectively, with radiation-induced tumor necrosis and fibrosis.

## 1. Introduction

Soft tissue sarcomas (STS) is a rare and highly heterogeneous tumor entity requesting a multidisciplinary approach [1]. The treatment of localized tumors is based on wide monobloc excision with negative margins in reference centers, mostly combined with perioperative radiotherapy (RT) [2]. With similar oncologic outcomes, image-guided preoperative RT is increasingly used compared to postoperative RT—with superior long-term functional outcomes despite a higher but temporary risk of wound complication [3,4,5]. On the surgical specimen, RT induces necrosis, cystic change, hemorrhage, hyalinization, and fibrosis. Histologic response was categorized [6], and a complete response is associated with a major prognostic impact [7]. In correlation, histopathological changes may influence dimension-based assessments of response significantly [8,9,10], even if tumor volume increase does not result in a worse outcome for patients [11]. Specific imaging recommendations, including multiparametric sequences, have been proposed [12]. Although not applicable to all STS subtypes, necrosis has been recognized as a treatment response indicator in osteosarcomas [13,14]. Additionally, alternative histological parameters such as fibrosis or cellularity can potentially be used for response assessment after neoadjuvant treatment [15,16]. RT-induced hyalinization/fibrosis was recently evidenced as a significant independent favorable predictor for relapse-free survival and overall survival [6]. Non-invasive tumor response assessment may impact management in selected patients.

Magnetic Resonance Imaging (MRI) remains the imaging method of choice for the identification, characterization, local staging, and post-treatment follow-up of STS patients [17]. However, morphologic and size criteria based on conventional MRI sequences pre- and post-contrast enhancement cannot reliably assess histologic response [8,18,19,20]. Diffusion-Weighted Imaging (DWI) and Dynamic Contrast-Enhanced (DCE) imaging have also been used for neoadjuvant therapy response assessment in patients with STS [15,16,21,22,23,24,25] and an increase in Apparent Diffusion Coefficient (ADC) values following neoadjuvant therapy has been reported [25,26]. Despite these investigations on the role of advanced MRI for the evaluation of STS, the potential of Magnetic Resonance (MR) findings in predicting histological response after neoadjuvant therapy has not yet been investigated.

We hypothesize that pretherapeutic advanced MRI can predict histological changes after RT in patients with STS.

## 2. Materials and Methods

### 2.1. Population

Among the patients included in the Osteoarticular Tumor Characterization by Advanced Imaging prospective trial (TUMOSTEO, NCT02895633), we considered the ones who benefited from preoperative RT against limb STS from 2010 to 2018. The indication of RT was validated during the institutional multidisciplinary STS board. Our center is part of the French reference network (NetSarc). TUMOSTEO was approved by the Ethical Committee (2009-a0068-49/09.09.07). Due to the anonymized and retrospective data analysis from a prospective study previously approved, an institutional review board exemption was granted. Written informed consent was obtained from all participants in this study.

### 2.2. Acquisition Technique

All images were acquired with a 1.5T MR system (Signa HDxt, General-Electric Healthcare, Milwaukee, WI, USA) with coils adapted to tumor location and patient body habitus before RT. The conventional acquisition protocol included a T1-weighted fast spin-echo (FSE) sequence, a T2-weighted FSE fat-saturated sequence in at least two different orthogonal planes, and a T1-weighted fat-saturated after the injection 0.2 mL/kg of a gadolinium-based contrast medium (gadobenate dimeglumine, MultiHance, Bracco Diagnostics, Monroe Township, NJ, USA). The contrast agent was injected into a peripheral vein at a rate of 0.5 mL/s using an injection pump (Spectris Solaris EP, Medrad Inc, Indianola, PA, USA).

Acquisition parameters were adapted to the anatomy at the tumor location. Field of view (FOV), slice thickness, and gap varied from 100 to 400 mm, 3.5 to 5 mm, 0.5 to 3 mm, respectively. The matrix size varied from 224 × 256 to 416 × 352. On T1-weighted sequences, the following acquisition parameters were used: repetition time (TR), 200–600 ms; echo time (TE) 2–17 ms; number of excitations (NEX) 1–4; bandwidth 13–50 kHz; echo train length (ETL) 2 to 4. On T2-weighted sequences, the following acquisition parameters were used: TR 3500–10,000 ms; TE 48–77 ms; NEX 1–4; bandwidth 13–42 kHz; ETL 10–23.

A single-shot pulsed gradient spin echo DWI sequence with echo-planar imaging (EPI) read-out was performed in three imaging axis directions with chosen b values of 0 and 600 s/mm^2^. DWI acquisitions were obtained before contrast media injection. The acquisition parameters used were TE minimal; TR 5000 ms; NEX 6; bandwidth 250 Hz; slice thickness 6 mm; gap 0; matrix 128 × 80. FOV and slice thickness were adapted to the patient’s anatomy and tumor size; the total acquisition time was 1 min 40 s.

DCE perfusion was performed with an optimized time-resolved 3D imaging of contrast kinetics with subtraction mask (TRICKS) using the following acquisition parameters: TE minimal; flip angle 30°; NEX 0.69–1; matrix 128 × 128 to 160 × 128; FOV 198 × 140 to 396 × 280 mm; bandwidth 50 kHz and thickness 3–5 mm. Twenty successive 40-slice volumes centered on the lesion were acquired with a 10-s inter-volume delay. The contrast injection started 15 to 35 s after the beginning of the acquisition, depending on the tumor location.

### 2.3. Image Post-Processing and Analysis

#### Functional MRI Analysis

Image post-processing of DWI and DCE was performed using a dedicated workstation (OleaSphere V. 2.3, Olea Medical SA^®^, La Ciotat, France) by a radiologist with four years of clinical experience with MRI, blinded to histological data. DWI and DCE data were automatically post-processed to calculate ADC (10^−3^ × mm^2^/s) and time-to-signal intensity curves, respectively.

The following formula was used for ADC calculation:ADC=∑I=1n−ln⁡(SiS0)bi
where *bi* = diffusion gradient value, *S*0 = signal intensity of the first image, and *Si* = signal intensity.

The slice demonstrating the largest tumor diameter was selected for analysis.

On ADC functional maps, the reader placed a free-form elliptical region of interest (ROI) on the solid tumor area that presented the lowest ADC value, based on the study of Bonarelli et al. [27]. The mean ADC value of the pixels in this ROI was considered to represent the minimal ADC of the tumor. The mean ADC value of skeletal muscle was obtained by placing another free-form elliptical ROI on normal-appearing muscle adjacent to the tumors evaluated of similar size (Figure 1). The tumor ADC ratio was calculated by dividing minimal tumor ADC by muscle ADC values.

DCE AUC functional maps were evaluated, and the reader placed a free-form elliptical ROI on the tumor area most intensively enhancing and on a nearby artery. A second ROI of the corresponding size was placed on adjacent normal-appearing striated muscle. Tumor-to-muscle ratios were obtained by dividing tumor AUC by muscle AUC values.

ADC and DCE AUC functional maps were analyzed in association with conventional morphologic images to avoid including areas of necrosis (non-enhancing tumor zones), areas of flow void (intratumoral tubular images of signal void) calcification (amorphous hypointense foci in all sequences) and normal tissue in tumor ROIs.

### 2.4. Radiotherapy

Target volumes and organs at risk were defined on CT before surgery in all patients. MR images were co-registered to construct the gross target volume. Clinical target volumes included a geometric expansion of 1.5 cm laterally and 4 cm longitudinally to encompass microscopic disease areas. The planning tumor volume included clinical target volumes plus an additional geometrical margin of 1 cm to cope with patient set-up errors. A 3-dimensional conformal external beam irradiation was delivered with a dose of 45–50.4 Gy in 25–28 daily fractions over 5 weeks.

### 2.5. Histopathological Analysis

The surgical specimens were analyzed by pathologists specialized in the evaluation of STS. Specimens were fixed in 4% formalin for at least 24 h. Representative tumor sections were obtained considering tumor heterogeneity, including at least one section per centimeter of the largest tumor diameter.

Microscopic analysis was made on 4 μm thick slices stained with hematoxylin and eosin. ST’s grading was performed according to the FNCLCC (Fédération Nationale des Centres de Lutte contre Le Cancer) system. Grade I and II lesions were considered to be low grades, whereas grade III lesions were considered to be high grades. The percentage of fibrosis (hypocellular areas with deposition of dense amorphous palely eosinophilic material and/or abundant collagenous matrix with sparse fibroblasts) and post-treatment necrosis (defined as a non-viable tumor area of necrotic tumor cells—anuclear “ghost cells”—in a background of neutrophilic infiltrates, degenerated cells, and extracellular nuclear debris). According to the EORTC-STBSG recommendation to assess tumor response, we categorized and reported the necrotic response grade into two classes: <50% (categories A–D) and ≥50% (category E). Tumor cellularity was also evaluated by quantifying the nuclear-to-stromal ratio, defined as the percentage of lesion nuclei to stromal tissue present. These percentages were calculated semi-quantitatively on every tumor glass slide and were averaged considering all the slides. Fibrosis was categorized into three grades: Grade I was considered less than 25% fibrosis, grade II between 25 and 75% fibrosis, and grade III more than 75% fibrosis after histologic analysis.

### 2.6. Statistics

Quantitative variables were expressed as mean ± standard deviation (range) or median, interquartile range (IQR) according to the normality of the distribution assessed by the Shapiro–Wilk test. Qualitative parameters were expressed as frequency and percentage. The relationship between MRI and quantitative histological parameters was investigated with the Spearman correlation coefficient. ADC values were compared according to the histologic necrosis rate (more or less than 50%) with a Mann–Whitney U test and according to tumor fibrosis grade with a Kruskal–Wallis test. Receiver Operating Characteristic (ROC) curves were used to evaluate the performance of DWI in identifying tumor necrosis. The significance level was set at 0.05. Statistical analyses were performed using SAS software, version 9.4 (SAS Institute Inc., Cary, NC, USA). Survival analyses were computed from the end of the radiotherapy to death, whatever the cause or last follow-up. It was estimated using the Kaplan–Meier method and compared with the Log-rank test. For progression-free survival, the event was local recurrence, metastasis recurrence, or death, whichever occurred first.

## 3. Results

### 3.1. Population

When we queried the TUMOSTEO database, 969 patients were included. We retrieved 31 patients who met the inclusion criteria. Two patients were excluded as one surgical resection was not performed, and in the other, advanced MRI sequences were not available. Among the 29 qualified patients, there were 15 women and 14 men (M/F sex ratio = 0.9) with a mean age of 61 ± 18 (19–84) years. The mean delay between preoperative RT and surgery was 60.5 ± 33.0 (32–172) days. The histological subtypes of the tumor studied are presented in Table 1. There were nine grade I (31%), 17 grade II (59%), and three grade III (10%) tumors. After RT, 20 patients had less than 50% necrosis (69%), whereas nine (31%) had more than 50% necrosis. In the latter patients, there was one grade I (11%), six grade II (67%), and two grade III (22%) tumors. Concerning fibrosis, six patients were considered grade I (21%), 11 grade II (38%), and 12 grade III (41%). The median cellularity in the tumors studied was 20 (IQR from 40 to 50) %. These data are reported in Table 2.

### 3.2. Clinical Outcome

The median follow-up was 66 months, with an interquartile range from 31 to 89 months. Six patients (21%) presented local recurrence.

The progression-free survival at 60 months was 55%, 95%CI 36–71% with a significant difference according to necrosis percentage (*p* = 0.010): 70%, 95%CI 45–85% for patients with necrosis < 50% and 22%, 95%CI 3–51% for patients with necrosis ≥ 50% (Figure 2) The overall survival at 60 months was 69%, 95%CI 49–82 without significant difference according to necrosis percentage (*p* = 0.139): 75%, 95%CI 50–89% for patients with necrosis <50% and 55%, 95%CI 20–80% for patients with necrosis ≥ 50% (Figure 2).

### 3.3. DWI and Histopathological Analysis

The median ADC value in the STS studied was 134.3 (101.7–167.9) × 10^−3^ mm^2^/s. The median tumor-to-muscle ADC ratio was 1.00 (0.77–1.55). There was a significant correlation between the ADC value and the post-treatment tumor necrosis percentage (*p* = 0.013) with a fair Spearman correlation coefficient (0.47). There was also a statistically significant correlation between the tumor-to-muscle ratio ADC and the post-treatment necrosis percent (*p* = 0.011) with a 0.48 Spearman correlation coefficient. Among the 20 tumors with less than 50% necrosis after RT, the median ADC value was 120.3 (97.7–161.6) × 10^−3^ mm^2^/s (Figure 3), while in tumors with more than 50% necrosis, the mean ADC value was significantly higher: 202.0 (160.9–243.6) × 10^−3^ mm^2^/s (*p* = 0.020) (Figure 4). For tumor-to-muscle ratio ADC, the difference was also significant with median ratios of 0.93 (0.72–1.25) and 1.55 (1.18 to 1.81) for tumors with under and over 50% of histologic necrosis (*p* = 0.045).

ROC analysis indicated that the ADC value of 161 × 10^−3^ mm^2^/s yielded a 95% sensitivity and a 55% specificity for the identification of tumors with more than 50% tumor necrosis ratio after RT (Figure 5). Similarly, a 1.4 tumor-to-muscle ADC ratio cutoff yielded an 85% sensitivity and a 55% specificity.

ADC values were not significantly associated with tumor cellularity and fibrosis grade (*p* = 0.070 and *p* = 0.603, respectively). Additionally, there was no statistical difference in the ADC values between patients who presented local recurrence and those who did not (*p* = 0.94).

### 3.4. DCE Perfusion and Histopathological Analysis

The median tumor-to-muscle AUC ratio in the tumors evaluated was 6.12 (3.52–7.24). MRI parameters are reported in Table 3. Tumor-to-muscle AUC ratios were associated with histologic fibrosis (*p* = 0.036) with a fair Spearman correlation coefficient (0.41). Tumor-to-muscle AUC ratios in STS with fibrosis grades I, II, and III were 2.65 (2.51–2.69), 5.74 (3.52–6.26), and 6.64 (5.13–8.54), respectively. Tumor-to-muscle AUC ratios were not significantly associated with post-RT necrosis (*p* = 0.627) nor cellularity (*p* = 0.502).

## 4. Discussion

As neoadjuvant therapy may potentially improve survival rates and facilitate surgical resection, we aimed to predict the tumor response to neoadjuvant therapy with functional MR techniques and identify which tumors could benefit from it. We explored standardized clinical data and images collected from a large prospective clinical trial. As expected, we observed a significant impact of RT-induced necrosis on PFS. However, the number of patients reported is insufficient to observe a correlation with overall survival. The main objective of our study was to search for baseline MRI parameters correlated with the RT response. There was a statistically significant correlation between the ADC values and the post-treatment tumor necrosis ratio (*p* = 0.013). Initial ADC values over 161 × 10^−3^ mm^2^/s presented a high sensitivity (95%) to predict more than 50% histologic necrosis ratio after neoadjuvant therapy. Although prior studies have demonstrated a correlation between ADC values and histologic necrosis, the potential of DWI in predicting histologic necrosis has not yet been reported [16]. The prediction of tumor histologic necrosis might be related to the fact that ADC values correlate with tumor biological aggressiveness and may reflect more than the treatment effect on tumor cells [28]. Indeed, factors such as extracellular microenvironment, alterations in cell membrane integrity, or water permeability, reflecting cellular pleomorphism, the presence of a myxoid component, and microscopic necrosis may impact ADC values [28,29,30,31,32]. Aggressive lesions tend to develop a complex tissue environment with highly permeable neovessels, which may be more sensitive to RT [33,34]. Although further investigations are needed to confirm these findings, DWI with ADC analysis could be an additional parameter to be considered in the evaluation of RT response in patients with STS.

The use of DWI with ADC analysis has been advocated for soft-tissue tumor characterization and recurrence detection [26,35,36,37,38]. This technique is widely available and can be easily integrated into a tumor characterization imaging workup. Lower ADC values are associated with malignancy and tumor aggressiveness, and in this regard, our results might seem contradictory at first sight. However, only STS were evaluated in this study, while prior studies on soft-tissue tumor characterization included benign and malignant lesions of various histologic subtypes [26,27,31,37,38]. Thus, the importance of ADC analysis for the evaluation of soft-tissue tumors might be two-fold, first for the differentiation between benign and malignant tumors and second for the prediction of neoadjuvant treatment response. Finally, AUC in DCE MRI significantly increased with the degree of histologic fibrosis (*p* = 0.036), which has been linked to hypoxia and tumor immunity with potential implications for patient management [39]. Previous studies suggested that fibrosis was associated with favorable outcomes [6] and was higher in patients with a good histologic response following preoperative RT [16]. None of the parameters evaluated presented an association with STS cellularity.

Several limitations of this study have to be acknowledged. First, our study was conducted in a single center with a limited number of patients due to the low prevalence of STS and the strict inclusion criteria. We evaluated a population of heterogeneous tumors, which probably influenced the prognosis performance of our parameters. Although imaging techniques such as PET CT could be interesting to compare with MRI, MRI was the only imaging technique that was systematically performed at baseline and follow-up in our data set. The correlation between ADC values and conventional histologic criteria for a beneficial neoadjuvant treatment would not be evaluated due to the small percentage of treatment responders in the study population [8,16]. A single reader performed image post-processing. Given the preliminary nature of this study and that various prior studies have shown excellent inter-observer agreement of ADC measurements, we believe the impact of this limitation to be negligible [15,16,21,25,27]. Only monoexponential DWI was evaluated in this study. Bi-exponential intra-voxel incoherent motion, diffusion tensor imaging, and diffusion kurtosis imaging could improve the performance of DWI for STS characterization and were not evaluated [40,41,42,43,44,45]. Deep learning-based radiomics models could improve tumor characterization or patient treatment follow-up and were also not evaluated in this study [46,47]. Finally, the impact of DWI on patient management and overall survival was not directly assessed in this study.

We must also report certain limitations and challenges associated with the use of functional imaging techniques, such as technical variations in measurement related to variable anatomical regions and magnetic field heterogeneity; the great histological variations in soft tissue sarcomas, the parameters evaluated could vary in MR scans from different vendors; finally, it is hard to have a direct correlation between the tumor area evaluated on histology and the areas targeted for measurements in MR imaging post-processing.

## 5. Conclusions

ADC and DCE-derived AUC values might provide useful information to predict the necrosis ratio and tumor fibrosis after neoadjuvant therapy in patients with STS. Higher ADC values were associated with higher histologic necrosis ratios after RT, and ADC values over 161 × 10^−3^ mm^2^/s yielded 95% and were strongly associated with histological necrosis ratios over 50%. Although further confirmatory studies are still required, our results suggest that DWI and DCE MRI could provide non-invasive biomarkers for determining histological response after RT.

## Figures and Tables

**Figure 1 cancers-16-00878-f001:**
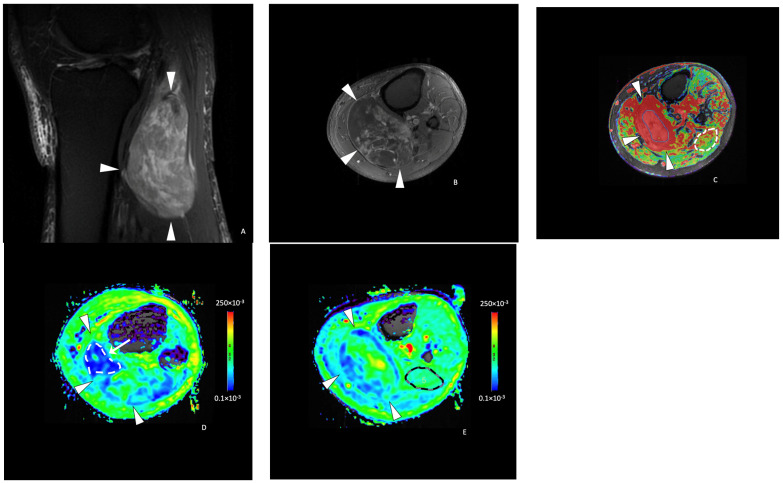
Functional imaging post-processing and analysis in a 69-year-old male with a leiomyosarcoma of the left popliteal fossa. Sagittal T2-weighted fat-saturated (**A**) and axial T1-weighted contrast-enhanced (**B**) MR images showing an ovoid mass (white arrowheads) with a heterogeneous hyperintense T2 signal and enhancement of the posterior compartment of the thigh in the popliteal fossa. (**C**) DCE (dynamic contrast-enhanced) AUC (area-under-the-curve) functional map in the axial plane showing a homogeneously enhancing tumor (arrowheads). Two free-form ROIs were placed. One on the area of highest tumor enhancement (green line) and another of the corresponding size in the muscle (white line). (**D**) Axial ADC (Apparent Diffusion Coefficient) functional map showing the same tumor (arrowheads). (**E**) An axial ADC functional map with an ROI of the corresponding size was drawn in the muscle (black line).

**Figure 2 cancers-16-00878-f002:**
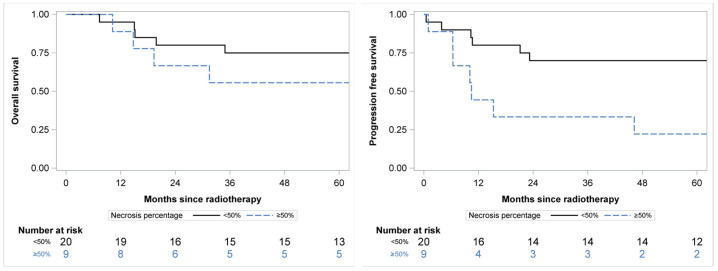
PFS (**left panel**) and OS (**right panel**) related to RT-induced necrosis level.

**Figure 3 cancers-16-00878-f003:**
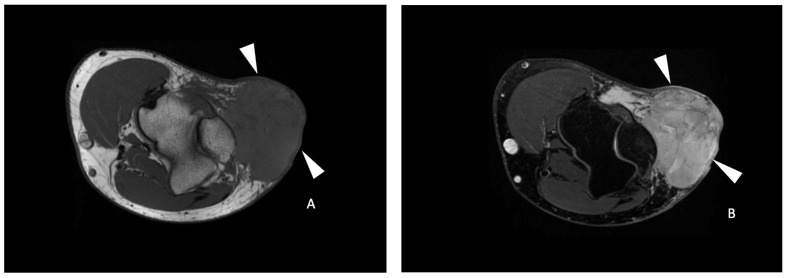
A 41-year-old male with a spindle cell sarcoma of the left elbow. Axial T1-weighted (**A**), axial T2-weighted fat-saturated (**B**) and axial T1-weighted contrast-enhanced (**C**) MR images showing an ovoid mass (white arrowheads) with a homogeneous hypointense T1 and hyperintense T2 signal with a heterogeneous enhancement with a central necrotic portion (arrow in (**C**)) in the superficial soft tissue, with close contact with the aponeurosis and the olecranon process. (**D**) Axial ADC functional map of the same tumor. A free-form ROI (white line) delineates the tumor area with the lowest ADC identified and was positioned within it, yielding a mean ADC of 92 × 10^−3^ mm^2^/s (<161 × 10^−3^ mm^2^/s). A free-form ROI (green line) of the corresponding size was positioned in the muscle. The tumor necrosis percentage after neoadjuvant therapy was 0%, corresponding to a poor response to treatment.

**Figure 4 cancers-16-00878-f004:**
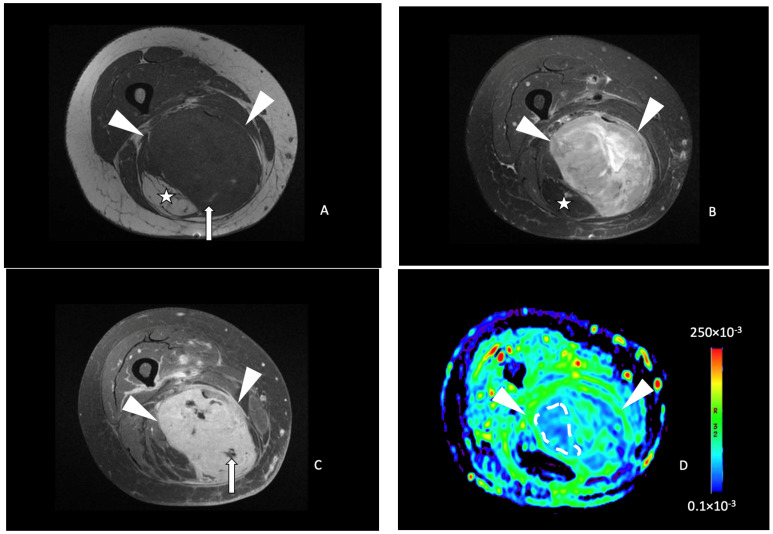
A 63-year-old female with a liposarcoma issue of the left thigh. Axial T1-weighted (**A**), T2-weighted fat-saturated (**B**), and T1-weighted contrast-enhanced (**C**) MR images showing a large heterogeneous tumor (arrowheads) of the posterior compartment of the thigh, arising from the semitendinosus and semimembranosus muscles containing hemorrhagic components (hyperintense signal T1, arrow in (**A**), a fat portion on its medial side (star in (**A**,**B**)) and small areas of intratumoral necrosis (arrow in (**C**)). (**D**) Axial ADC functional map showing a tumor with high ADC value measured at 288 × 10^−3^ mm^2^/s in the white dashed lined ROI (>161 × 10^−3^ mm^2^/s). The tumor necrosis percentage after neoadjuvant therapy was 60%.

**Figure 5 cancers-16-00878-f005:**
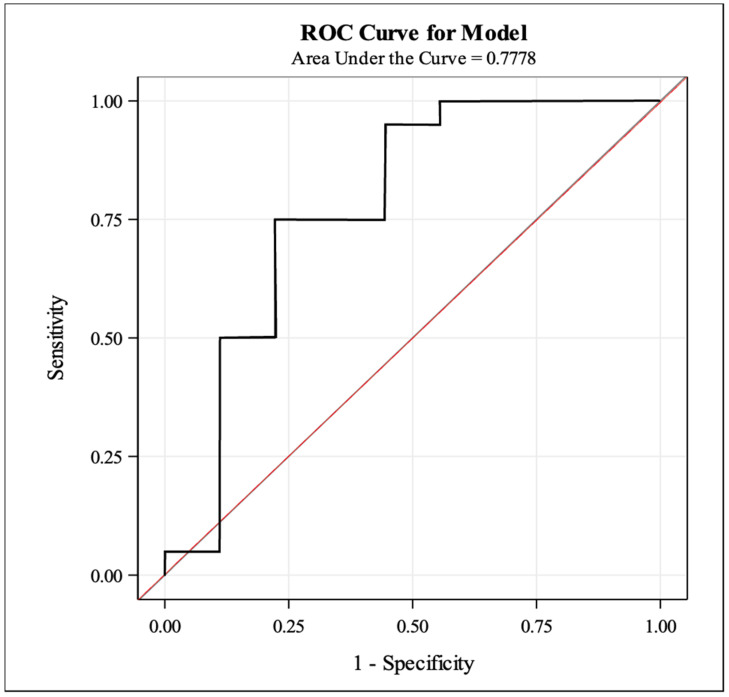
Sensitivity and specificity performance of ADC value: The graph shows the ROC analysis of ADC values for the identification of histologic necrosis ratio higher than 50%. The red line illustrates random classification.

**Table 1 cancers-16-00878-t001:** Demographic and histologic characteristics.

	Necrosis Percentage <50% (*n* = 20)	Necrosis Percentage ≥50% (*n* = 9)	*p*-Value
Age, mean ± SD (range)	59 ± 19 (19–83)	64 ± 10 (46–84)	0.65
Male Female	10 (50%) 10 (50%)	4 (44.4%) 5 (55.6%)	1
Histologic diagnosis			NC
- Myxoid fibrosarcoma	3 (15%)	2 (22.2%)	
- Undifferentiated sarcoma	4 (20%)	1 (11.1%)	
- Liposarcoma	2 (10%)	2 (22.2%)	
- Myxofibrosarcoma	3 (15%)	0	
- Dedifferentiated liposarcoma	1 (5%)	1 (11.1%)	
- Spindle cell sarcoma	1 (5%)	2 (22.2%)	
- Unclassified pleomorphic sarcoma	2 (10%)	0	
- Leiomyosarcoma	1 (5%)	0	
- Malignant peripheral nerve sheath tumor	1 (5%)	0	
- Pleomorphic liposarcoma	1 (5%)	0	
- Pleomorphic undifferentiated sarcoma	0	1 (11.1%)	
- Synovial sarcoma	1 (5%)	0	
FNCLCC grade			0.844
1	7 (35%)	2 (22.2%)	
2	11 (55%)	6 (66.7%)	
3	2 (10%)	1 (11.1%)	

SD = standard deviation; FNCLCC = Fédération Nationale des Centres de Lutte contre Le Cancer (French Federation for Cancer Centers). NC: Not Computed.

**Table 2 cancers-16-00878-t002:** Histological response following RT.

Cellularity percentage, median (IQR)	20 (40–50)
Necrosis percentage, median (IQR) <50% ≥50%	20 (0–60) 20 (69.0%) 9 (31.0%)
Fibrosis grade I grade II grade III	6 (21%) 11 (38%) 12 (41%)

RT = radiotherapy; IQR = Interquartile range.

**Table 3 cancers-16-00878-t003:** MRI parameters before RT.

ADC, median (IQR)	134.3 (101.7–167.9)
Tumor-to-muscle ADC ratio, median (IQR)	1.00 (0.77–1.55)
AUC, median (IQR)	70.108 (34.778–158.224)
Tumor-to-muscle AUC ratio, median (IQR)	6.12 (3.52–7.24)

MRI = Magnetic Resonance Imaging; RT = Radiotherapy; ADC = Apparent Diffusion Coefficient; IQR = Interquartile Range; AUC = Area Under the Curve.

## Data Availability

Research data are stored in an institutional repository and will be shared with the corresponding author upon request. The raw data supporting the conclusions of this article will be made available by the authors on request.

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
