# Peer review of "Evaluation of the Prognostic Value of Pretherapeutic Magnetic Resonance Imaging in Predicting Soft Tissue Sarcoma Radiation Response: A Retrospective Study from a Large Institutional Sarcoma Imaging Database"

_cancers, 2024, doi:10.3390/cancers16050878_

Round 1

Reviewer 1 Report

Comments and Suggestions for Authors

Radiation therapy-induced hyalinization/fibrosis has recently been shown to be an important independent predictor of complete response to neoadjuvant radiotherapy (RT) and survival in patients with soft tissue sarcoma (STS). The authors analyzed 29 patients diagnosed with STS who underwent neoadjuvant radiation therapy for STS of the extremities, recruited from a single center prospective clinical imaging database. Tumor apparent diffusion coefficients (ADC) and area under the time-intensity perfusion curve (AUC) were shown to be positively correlated with tumor necrosis rate after treatment (P = 0.013). Median ADC values were lower in patients with less than 50% necrosis and higher in patients with more than 50% necrosis (P = 0.020).

1. Very small sample, especially considering the diversity of histological types of sarcoma, as indicated in Table 1

2. Does it make sense to distinguish between Spindle cell sarcoma and Unclassified spindle cell sarcoma? In general, for the purposes of the study, the histological grading is not very important, however, it is interesting to discuss, for example, which histological samples fell into the group with necrosis of more than 50% and less than 50%.

3. I suggest the authors add a table and compare 2 groups: with necrosis of more than 50% and less than 50% according to the indicators given in Table 1.

4. What is the point in assessing non-invasively, if in the future there is surgical treatment and you can work directly with histology?

5. Only necrosis was assessed, but not survival rates depending on it, right? Why not add survival analysis?

Author Response

Dear reviewer 1; Thank you very much for your comments that will help us to clearly improve the quality of the manuscript. 

1- We acknowledge that our cohort is small given the rarity of the pathology explored - which also includes more than 150 histological and molecular subtypes. All the patients reported in our study were indeed retrospectively retrieved from a single-center prospective interventional study that recruited 969 patients with bone or peripheral soft-tissue tumor and evaluated multiple functional imaging methods (perfusion, diffusion, spectroscopy) for initial benign/malignant characterization whatever their management (NCT02895633). Among all the patients included in TUMOSTEO between 2010 and 2020, only 31 were managed with preoperative RT and an appropriate clinical/radiological followup. Preoperative RT has become more widespread in our center since 2016.

2- Thank you, you are right. We merged them in the new Table 1.

3 & 5- Thank you for this opportunity of added value. Table 1 was modified after restratifying the patients accordingly. Moreover, we performed a new statistical analysis to evaluate the correlation between necrosis and clinical outcome as requested in point #5. Meanwhile we adapted the follow-up of the patients. The overall survival at 60 months was 69%, 95%CI 49-82 without significant difference according to necrosis percentage (p= 0.139) : 75%, 95%CI 50-89% for patients with necrosis<50% and 55%, 95%CI 20-80% for patients with necrosis≥50% The progression free survival at 60 months was 55%, 95%CI 36-71% with significant difference according to necrosis percentage (p= 0.010) : 70%, 95%CI 45-85% for patients with necrosis<50% and 22%, 95%CI 3-51% for patients with necrosis≥50%.

4- As neoadjuvant therapy may potentially improve survival rates and facilitate surgical resection, the idea was to predict the tumor response to neoadjuvant therapy with functional MR techniques and identify which tumors could benefit from it.

Reviewer 2 Report

Comments and Suggestions for Authors

The manuscript investigates the potential of advanced MRI techniques, specifically DWI and DCE MRI, in predicting histological response after neoadjuvant therapy in patients with soft tissue sarcomas (STS). The study, conducted on a cohort from the Osteoarticular Tumor Characterization by Advanced Imaging trial, explores the correlation between imaging parameters and histopathological changes. Their results indicated that higher Apparent Diffusion Coefficient values were associated with increased histologic necrosis after radiotherapy, showing strong predictive capability. The findings suggest that DWI and DCE MRI could serve as non-invasive biomarkers for assessing histological response after neoadjuvant therapy in STS patients. To provide a more comprehensive understanding, the following analysis could be incorporated:

1) Have authors evaluated the correlation between imaging parameters and conventional histological criteria used to assess neoadjuvant treatment response?

2) Have authors thought to integrate or compare other imaging techniques with MRI?

3) Any potential limitations and challenges associated with the use of functional imaging techniques?

Author Response

Dear reviewer 2; Thank you very much for your comments that will help us to clearly improve the quality of the manuscript. 

1) Have authors evaluated the correlation between imaging parameters and conventional histological criteria used to assess neoadjuvant treatment response?

First of all, we restratified the patients according to the necrosis rate on the specimen (≥50% vs. <50%); Due to the sample size, it was not possible to analyze the tumor response in finer strata as suggested by the EORTC-STBSG 2016 recommendations (Wardelmannet al.). For memory :
A. No stainable vital tumor cells
B. Single stainable tumor cells or small cluster (overall < 1%)
C. ≥1% <10% stainable vital tumor cells
D. 10% <50% stainable vital tumor cells
E. ≥50% stainable vital tumor cells
We therefore pooled classes A, B, C, D vs. E.). We propose a new table 1 accordingly.

Then, we updated the followup and performed a correlation analysis between tumor response (necrosis) and classic clinical endpoints.
The overall survival at 60 months was 69%, 95%CI 49-82 without significant difference according to necrosis percentage (p= 0.139): 75%, 95%CI 50-89% for patients with necrosis<50% and 55%, 95%CI 20-80% for patients with necrosis≥50%
The progression free survival at 60 months was 55%, 95%CI 36-71% with significant difference according to necrosis percentage (p= 0.010): 70%, 95%CI 45-85% for patients with necrosis<50% and 22 %, 95%CI 3-51% for patients with necrosis≥50%.
The imaging criteria are listed in Table 3 and an analysis according to the necrosis rate was done and reported in Figure 4. With the data we have, it is not possible to evaluate other imaging parameters. All data collected in the prospective TUMOSTEO trial were analyzed.

2) Have authors thought to integrate or compare other imaging techniques with MRI?

In the TUMOSTEO prospective study (NCT02895633), the following imaging techniques were performed: contrast enhanced ultrasound, Low dose CT perfusion, Magnetic resonance perfusion, Diffusion weighted imaging, Magnetic resonance proton spectroscopy.

The following parameters were collected: 

  • Contrast enhancement in echography: yes or no [Time Frame: baseline]
  • Contrast enhancement curve (absent, slow or fast) in echography [Time Frame: baseline]
  • Maximal intensity of enhancement peak in echography [Time Frame: baseline]
  • Mean transit time of enhancement in echography [Time Frame: baseline]
  • Gradient of contrast enhancement curve in echography [Time Frame: baseline]
  • Vascularization (dynamic analysis) in CT scanner [Time Frame: baseline]
      • 0: absent
      • 1: low (< 20UH)
      • 2: > 20 UH without new blood vessels
      • 3: > 20 UH and/or with new blood vessels
  • Contrast enhancement curve (absent, slow or fast) in CT scanner [Time Frame: baseline]
  • Maximal intensity of enhancement peak in CT scanner [Time Frame: baseline]
  • Mean transit time of enhancement in CT scanner [Time Frame: baseline]
  • Gradient of contrast enhancement curve in CT scanner [Time Frame: baseline]
  • Contrast enhancement curve (absent, slow or fast) in MRI [Time Frame: baseline]
  • Maximal intensity of enhancement peak in MRI [Time Frame: baseline]
  • Mean transit time of enhancement in MRI [Time Frame: baseline]
  • Gradient of contrast enhancement curve in MRI [Time Frame: baseline]
  • Choline peak in magnetic resonance spectroscopy [Time Frame: baseline]
    • Choline presence was defined as a clear metabolite peak at 3.2 ppm
  • Apparent diffusion coefficient (ADC) in MR diffusion weighted imaging [Time Frame: baseline]
    • ADC value in mm2/s
  • Area under the perfusion curve in MRI [Time Frame: baseline]
  • Perfusion curve gradient in MRI [Time Frame: baseline]
  • Perfusion time-to-peak in MRI [Time Frame: baseline]

Although imaging techniques such as PET CT could be interesting to compare with MRI, in our data set MRI was the only imaging technique that was systematically performed at baseline and during the followup with robust parameters. This point can be added as a limitation to this work. 

3) Any potential limitations and challenges associated with the use of functional imaging techniques?

We acknowledge that there are a few. In our opinion the most important are: technical variations in measurement related to variable anatomical regions and magnetic field heterogeneity; the great histological variations in soft tissue sarcomas, the parameters evaluated could vary in MR scanners from different vendors, finally, it is hard to get a direct correlation between the tumor area evaluated on histology and the areas targeted for measurements in MR imaging post-processing.

The suggested modifications were incorporated in the new version of the manuscript (mostly in the "Discussion"). 

Round 2

Reviewer 1 Report

Comments and Suggestions for Authors

I have no further comments on the article; the authors have made the appropriate changes. I believe that in its present form the manuscript can be recommended for publication.